# Multi-Institutional Retrospective Case-Control Study Evaluating Clinical Outcomes of Foals with Small Intestinal Strangulating Obstruction: 2000–2020

**DOI:** 10.3390/ani12111374

**Published:** 2022-05-27

**Authors:** Sara J. Erwin, Marley E. Clark, Julie E. Dechant, Maia R. Aitken, Diana M. Hassel, Anthony T. Blikslager, Amanda L. Ziegler

**Affiliations:** 1Department of Clinical Sciences, College of Veterinary Medicine, North Carolina State University, Raleigh, NC 27607, USA; sjerwin@ncsu.edu (S.J.E.); meclark5@ncsu.edu (M.E.C.); anthony_blikslager@ncsu.edu (A.T.B.); 2Department of Surgical and Radiological Sciences, University of California-Davis, Davis, CA 95616, USA; jedechant@ucdavis.edu; 3Department of Clinical Studies, New Bolton Center, School of Veterinary Medicine, University of Pennsylvania, Kennett Square, PA 19348, USA; maitken@vet.upenn.edu; 4Department of Clinical Sciences, College of Veterinary Medicine, Colorado State University, Fort Collins, CO 80523, USA; diana.hassel@colostate.edu

**Keywords:** horse, colic, foal, ischemia, small intestinal strangulating obstruction, surgery

## Abstract

**Simple Summary:**

Lower survival rates have been reported in foals than adults with severe colic lesions obstructing blood flow to the small intestine, but this has not been compared directly. These survival rates are important to horse owners making medical decisions surrounding colic, for both the foal’s wellbeing and the owner’s finances. In this retrospective case-control study, hospital records of surgical colic cases were collected from five US academic referral hospitals to directly compare foal and adult survival following surgery for specific colic lesions. It was hypothesized that foals would exhibit lower survival than case-matched adults. This study was limited by incomplete medical and surgical records, relatively small sample size, and lack of long-term follow-up. Short-term survival in foals was not significantly different than in adults with comparable colic lesions and may have been partly driven by decision-making on the farm prior to referral. More optimism toward surgical treatment of foals with suspected SISO may be warranted.

**Abstract:**

Lower survival has been reported in foals than adults with small intestinal strangulating obstruction (SISO), but age-dependent outcomes have not been examined directly. Hospital records were collected from five US academic referral hospitals. It was hypothesized that foals would exhibit lower survival than case-matched adults. Foal cases 6-months-of-age or younger, and adult cases between 2- and 20-years-of-age were collected. Data revealed 24 of 25 (96.0%) foals and 66 of 75 (88.0%) adults that were recovered from surgery for SISO survived to hospital discharge. Sixteen of the total 41 (39.0%) foals studied were euthanized intraoperatively, whereas 30 of 105 (28.6%) adults were euthanized intraoperatively. Common lesions in foals that were recovered from surgery were volvulus (*n* = 13) and intussusception (*n* = 5), whereas common lesions in adults were volvulus (*n* = 25) and strangulating lipoma (*n* = 23). This study was limited by incomplete medical records, relatively small sample size, and lack of long-term follow-up. Unexpectedly, short-term survival tended to be higher in foals than adults and may have been partly driven by case selection prior to referral or surgery or decision-making intraoperatively. More optimism toward surgical treatment of foals with SISO may be warranted.

## 1. Introduction

Colic is the leading cause of death in adult horses, accounting for over 30% of deaths in horses from 1- to 20-years-of-age [1,2,3]. The most fatal form of colic in adult horses is strangulating obstruction [4], including strangulating lipomas of the small intestine where the reported mean age of affected horses is 19.2 years, and epiploic foramen entrapment of the small intestine where the reported mean age of affected horses is 9.6 years of age [5]. Reported rates of short-term survival from surgery to hospital discharge for these diseases range from 50–80% [5]. Alternatively, short-term survival rates for foals experiencing strangulating lesions of the small intestine, most commonly small intestinal volvulus, range from 27–50% [6]. While age-dependent outcomes have been examined in adult horses [7], survival rates have not been directly compared between foals and adults [6,8]. Reported long-term survival rates of foals that survived to hospital discharge following exploratory celiotomy ranges from 33–75%, depending on the lesion and the foal’s age at surgery [9]. Recently, reported survival rates for surgical correction of colic lesions in adult horses have risen to between 80–95% depending on the lesion, and horses have been shown to return to use more frequently than previously reported [10,11].

The decision to recommend referral for surgery is based on the horse’s apparent level of pain on presentation, vital signs (particularly the heart rate) [12,13], peritoneal fluid and lactate [14,15], the duration of colic symptoms, response to analgesia [12], evidence of reflux with nasogastric intubation, and findings on palpation per rectum [16]. The speed of referral is pivotal in optimizing survival rates of equids with small intestinal obstructions [17], with a 6-h duration of strangulating obstruction having a favorable prognosis, but a marked reduction in survival rates after a 12 h duration of strangulating obstruction [6,10]. The referral process is complicated in the foal due to varying degrees of visible or recognizable pain in young horses, and the inability to perform rectal palpation in the foal [8]. Abdominal radiographs and ultrasound are useful for detecting possible lesions in foals, but these modalities may not be commonly used in equine primary care practice, especially in the foal [6,8,18]. The communication process between the referring veterinarian, surgeon, and foal owner is further complicated by previous reports of lower survival rates in foals undergoing colic surgery compounded with overestimation of the cost of colic surgery, and whether or not the foal is insured [19]. Misconceptions and outdated literature may bias the way in which veterinarians present referral options to foal owners. Furthermore, an owner’s decision-making may be affected by their previous experiences with colic, or by shared stories from other horse owners [19].

Recent studies have reported increased survival rates for horses of all ages following colic surgery, and higher than expected return to use has been reported as well [10,11]. However, age-dependent outcomes following surgical management of colic have not been compared directly. This retrospective case-control study aimed to examine differences in clinical outcomes between foal and case-matched adult SISO patients, with the objective of generating knowledge pertinent to both clinicians and horse owners, to facilitate a dialogue regarding more accurate expectations prior to referral. 

## 2. Materials and Methods

### 2.1. Inclusion Criteria

Hospital records from five institutions were reviewed to find surgical SISO cases. These records were obtained from North Carolina State University, University of California, Davis, Colorado State University, University of Pennsylvania and The Ohio State University. Horses were included if SISO was confirmed during laparotomy, with adults defined as horses between 2- and 20-years-of-age, and foals defined as 6-months-of-age or younger. Records were collected for both horses that were recovered from surgery and those that were euthanized intraoperatively, though the cases euthanized intraoperatively were analyzed separately. Three adult cases recovered from surgery were matched to each foal case that was recovered from surgery. Attempts were made to match adult cases to foals by institution, year of surgery, type and location of lesion, and whether a resection was performed, based on the medical records available at each referral hospital (Figure 1). Epiploic foramen entrapments were excluded from analysis due to previous research indicating a much poorer prognosis compared to other SISO lesions [20]. Types of lesions matched were approximate, as some cases present with lesions not seen in the other age group. For example, a foal was included with SI strangulation secondary to an ascarid impaction (an uncommon impaction etiology in adult horses) and was matched to adults that also had strangulated SI secondary to impactions of feed material. Foals and adults with SISO euthanized intraoperatively were excluded from survival analyses and therefore not case-matched, but were used to examine possible differences in intraoperative decision-making between foals and adults with SISO.

### 2.2. Statistical Analyses

GraphPad Prism software was utilized for all statistical analyses. Age distributions were evaluated for normality using Shapiro–Wilk normality tests. Univariate analyses were performed to assess selected factors with survival. Factors assessed included age (foal or adult), sex, breed, lesion, and whether or not a resection was performed. In addition, the distribution of breed and of intraoperative euthanasia decisions was assessed in foals as compared to adults using simple logistic regression or Fisher’s exact tests when appropriate. *p* < 0.05 was considered significant.

## 3. Results

### 3.1. Age Distribution

The population of foal cases collected was not normally distributed by age (*p* = 0.0002) (Figure 2a). The foal population appeared to have a right skew (mean = 67.34 days-of-age, median = 42 days-of-age), with younger foals presenting more frequently than older foals, particularly foals under 60-days-of-age (Figure 2a). When broken down by intraoperative decision, the population of foals euthanized during surgery was normally distributed (*p* = 0.07) (mean = 65.34 days of age, median = 40 days of age), with a trend toward intraoperative euthanasia of younger foals to achieve a more normal distribution from an originally right-skewed distribution (Figure 2b). Similar to the foal population, the adult population was not normally distributed (*p* < 0.0001) but is bimodal in appearance, with older and younger adult horses presenting more frequently than middle-aged horses (*n* = 20 horses between 17 and 18 years of age, *n* = 22 horses between 6 and 8 years of age) (Figure 2c). The distribution of adults euthanized during surgery was not normally distributed and followed the same bimodal appearance of the full population (*p* = 0.02) (Figure 2d).

### 3.2. Breed Distribution

Some breeds presented more frequently with SISO, including Thoroughbreds, the warmblood breed group, Arabians, and Standardbreds. Of note, Standardbred foals were overrepresented in the total population when compared to the adult full population (*p* = 0.02), but were no longer overrepresented when comparing foals and adults that were recovered from surgery (*p* = 0.08). In total, 25 of 41 foals were recovered from surgery, including 9 Thoroughbreds, 6 Standardbreds, 4 warmbloods, 2 Arabians, 2 Andalusians, 1 Quarter Horse, and 1 Percheron. A total of 75 of the 105 adults were recovered from surgery, including 22 Thoroughbreds, 12 Quarter Horses, 10 warmbloods, 7 Arabians, 7 Standardbreds, 2 ponies of unspecified breed, 2 Morgans, 2 American Saddlebreds, 2 Tennessee Walking Horses, 1 Shire, 1 Percheron, 1 Quarab (Arabian-Quarter Horse cross), 1 Gypsy Cob, and 5 others of unspecified or unknown breed. Further information on breed distributions can be found in the Appendix A.

### 3.3. Lesion Distribution

Lesions that presented in foals that were recovered from surgery included small intestinal volvulus, mesenteric rents, inguinal hernias, intussusceptions, umbilical Richter’s hernias, ischemia secondary to an impaction, and strangulation by a mesodiverticular band.

Lesions that presented in adults recovered from surgery included small intestinal volvulus, mesenteric rents, inguinal hernias, omental rents, strangulating lipomas, gastrosplenic ligament entrapments, nephrosplenic ligament rents, and SI entrapment in the abdominal wall. Detailed case-matching by age and lesion is available in the Appendix A.

### 3.4. Survival Analysis

Sixteen of 41 (39.0%) foals and 30 of 105 (28.6%) adults were euthanized during surgery (*p* = 0.2; 95% CI = 0.3 to 1.4) (Figure 3a). Assessment of survival revealed that 24 of 25 (96.0%) foals and 66 of 75 (88.0%) adults that were recovered from surgery for SISO survived to hospital discharge (Figure 3b). Of the foals that survived to hospital discharge, 9 (37.5%) had resections, and 27 (40.9%) of the surviving adults had resections (*p* = 0.5; 95% CI 0.8 to 2.7). Of the 9 adults euthanized postoperatively, 5 (55.6%) had resections. The 1 foal that was euthanized postoperatively did not have a resection. When analyzing cases recovered from surgery, there was a trend toward higher survival in foals as compared to adults (*p* = 0.4; 95% CI 0.4 to 37.3) (Figure 3b). 

Long-term follow-up was available for 13 adults and 5 foals, showing that there was a 60% long-term survival in foals and 84.62% long-term survival in adults, though the cases available were too few for statistical analysis (Figure 4). Of those available for long term follow up, 2 foals and 2 adults were euthanized 1 month post-operatively due to the formation of adhesions.

### 3.5. Assessment of Factors Associated with Survival

Factors evaluated for their association with survival were age, breed, and lesion (Table 1). No significant risk factors were identified in the univariate analyses.

## 4. Discussion

There were no significant differences in the number of foals and adults euthanized at surgery, and a similar number of foals and adults underwent intestinal resection. The short-term survival of foals with SISO was higher but not significantly different than case-matched adult horses (96% vs. 88%, *p* = 0.4) despite a similar proportion of intra-operative euthanasia decisions and resections between the two age groups. This contrasts with previous studies examining foal survival rates [6,8,9], and is suggestive of a more favorable short-term prognosis for foals with surgical colic lesions as compared to prior reports.

Long-term follow-up was incomplete in this study, so no conclusions can be drawn regarding foal or adult prognoses past discharge from the hospital. This may be a target for future research, as updated long-term survival statistics would be of similar interest to horse owners and veterinarians.

This retrospective study examines data regarding survival of foals and adult horses with surgical colic lesions. Regardless of the pathophysiology involved in the lesion and the prognosis provided by the referral veterinarian or surgeon, horse owners must make their own decisions to either continue pursuit of treatment or elect for humane euthanasia. Human decision-making undoubtedly biases the survival outcomes, which would likely be different if there was no human intervention involved, and the sociological factors in the decision-making process of horse owners and veterinarians may not be ignored. 

The relatively higher rate of survival in foals may have been partly driven by case-selection prior to referral, or following initial workup in the referral hospital, but these data were not available as the clinical records evaluated were of horses that had owners that elected referral for further colic work-up. Many of the decisions to pursue referral and surgery are financially driven, regardless of the animals’ age or prognoses provided [19]. While it is known that surgeons expect foals to form intrabdominal adhesions more readily than adult horses [21,22,23,24,25], recent studies are showing this may not be the case [25,26]. Regardless, this knowledge is commonly held and may further bias the way surgeons view foal cases intraoperatively. It is also possible that fewer foals are referred because they are not yet proven as, for example, adult show horses in their intended discipline. There is literature focusing on primary care veterinarians’ decision-making around recommending referral [19,27] and on surgeons’ intraoperative decision-making [10], although this has not focused on decision-making in foal cases. Sociological research conducted in the UK and Europe focused on horse owners’ abilities to recognize serious colic symptoms and their views on referring a horse for further workup and possible surgery. Researchers found that most owners can recognize signs of colic in their horse, but the immediate actions taken were dependent on the extent of their experience with colic, and on their ability to identify symptoms of more severe cases [28]. The decisions to refer for further workup and possible surgery can also be impacted by other factors, including geographic region relative to a referral hospital, and whether horse owners view their horses more as companion animals (pets) or as working animals and a source of income [28]. Similar sociological research is lacking in the US and identifies an important gap in the knowledge that undoubtedly impacts colic survival rates [28,29]. Without adequate qualitative data from horse owners and primary care veterinarians in the US, this study cannot draw conclusions about the proportion of foals or horses that may have been euthanized after initial workup on the farm or at the hospital, nor what the leading reason was for decisions on euthanasia.

Furthermore, the inherent differences in geographical location likely had some impact on the demographics seen in this study. For example, Standardbred foals were overrepresented in one population, but it is worth noting that many of the Standardbreds included in the study were admitted to the University of Pennsylvania (UPenn), where this distribution is an adequate representation of the surrounding population, as there are many Standardbred racehorse breeding farms in the Pennsylvania area. UPenn tends to see fewer Standardbred adults compared to Thoroughbred adults. Anecdotally, this is because Standardbred horses are more likely to be sold to Amish families following their racing career, so they are less likely to have a show career post-racing than a Thoroughbred and fewer referral options as adults than other breeds, which leads to the abundance of Standardbred foals referred relative to a lack of Standardbred adults. 

In addition, differences in lesion types likely had some impact on the survival rates seen in this study. Foals and adult horses often experience lesions that are specific to certain age groups. For example, aged geldings are predisposed to developing pedunculated lipomas which encircle and strangulate sections of intestine, while foals do not develop these lipomas [30]. In contrast, foals more commonly experience ascarid impactions, an uncommon lesion in adult horses [31]. With inclusion of these age-specific lesions, efforts were made to case-match lesions that were comparable in pathophysiology. While these efforts did not produce perfectly matched control cases, it was possible to identify control lesions similar enough in severity and pathophysiology to be compared. For example, an adult with ischemic injury secondary to an impaction of feed material was compared to a foal with ischemic injury secondary to the ascarid impaction. While it has previously been noted that ascarid impactions have poorer outcomes than other types of impactions [31], surgical notes were referenced to validate similarity of lesion severity, though this is ultimately difficult to guarantee due to frequently incomplete surgical records.

Finally, the age range for enrolled adult colic cases was capped at 20-years-of-age to limit impact of age-related comorbidities on adult survival. Widening the age range for adult controls might have increased the number of adults that could have served as better controls in some instances. 

Overall, limitations of this study include a relatively small sample size and lack of follow up on the majority of cases, rendering long-term survival analyses and conclusions regarding long-term foal survival impossible.

## 5. Conclusions

The results of this study demonstrate no significant difference in short-term outcome of surgically treated SISO foals compared with adults. The clinical application of such findings supports more optimism toward surgical treatment of foals with SISO. First opinion veterinarians should be encouraged to discuss a surgical referral option for owners of foals with suspected SISO more favorably given these updated short-term survival rates. More research is required in this area, as more data and updated representative statistics regarding foal survival for referring veterinarians will allow foal owners to make the most appropriate decision for the foal’s wellbeing and allow for judicious use of the owner’s finances. Future work might focus on elucidating the factors that influence the decision-making of surgeons and foal owners.

## Figures and Tables

**Figure 1 animals-12-01374-f001:**
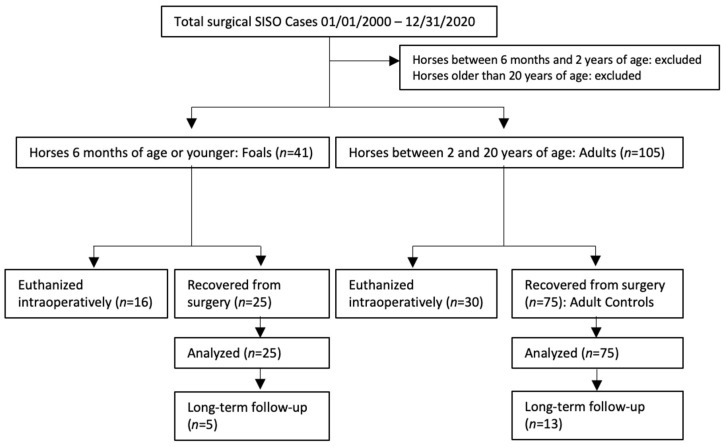
Case selection and inclusion flow chart.

**Figure 2 animals-12-01374-f002:**
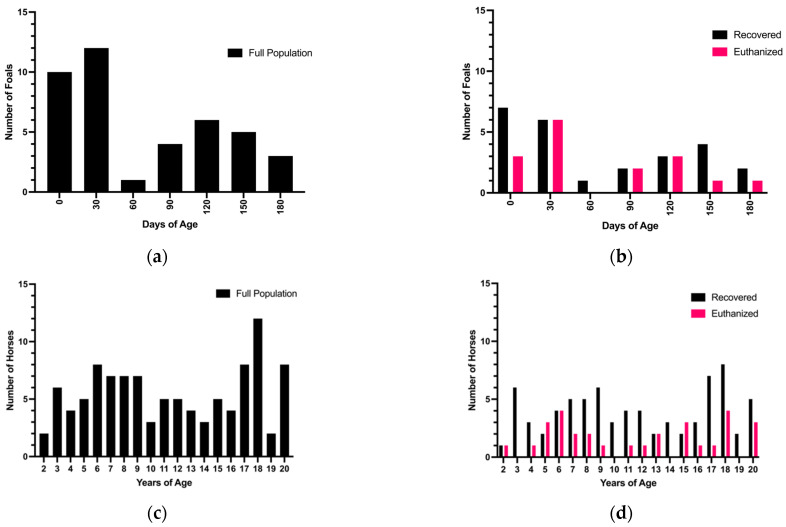
Age affects intraoperative decision in foals. (**a**) shows the frequency distribution of foals by days of age, binned by months. The full population of all foal cases collected was not normally distributed, and appears to be skewed right, with more younger foals presenting than older foals (mean = 67.34 days-of-age, median = 42 days-of-age) (Shapiro–Wilk normality test *p* = 0.0002, W = 0.8655). (**b**) When separated based on intraoperative decision, the population of foals euthanized during surgery was normally distributed, while the population of foals recovered from surgery was not (mean = 65.34 days of age, median = 40 days of age) (Shapiro–Wilk normality test *p*= 0.071, W = 0.8966 and *p* = 0.0021, W = 0.8490, respectively). (**c**) The full population of adult cases collected was not normally distributed, and appears to have a bimodal shape, with younger and older adults presenting more frequently than adult horses between 10 and 16 years of age (*n* = 20 horses between 17 and 18 years of age, *n* = 22 horses between 6 and 8 years of age) (Shapiro–Wilk normality test *p* < 0.0001, W = 0.9326). (**d**) When separated by intraoperative decision, both the population of adults that were euthanized in surgery and recovered from surgery were not normally distributed, (Shapiro–Wilk normality test *p* = 0.0175, W = 0.9127 and *p* = 0.0009, W = 0.9358, respectively).

**Figure 3 animals-12-01374-f003:**
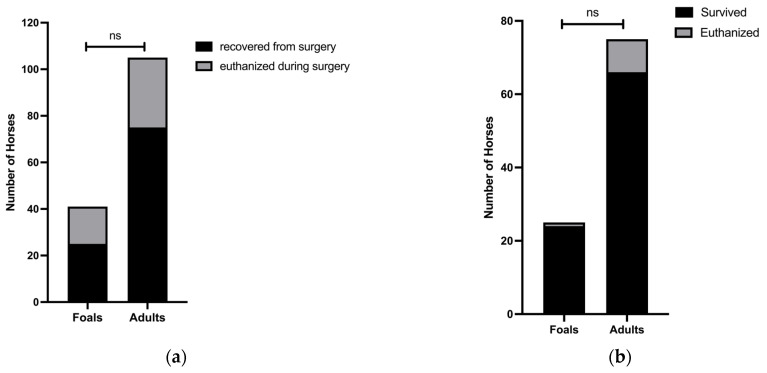
Similar proportions of foals and adults were euthanized intraoperatively, and short-term survival was higher in foals than adult case-controls. (**a**) Of 41 foals, 16 (39.0%) were euthanized intraoperatively, while 30 of 105 (28.6%) adults were euthanized intraoperatively. (**b**) Of 25 foals, 24 (96.0%) survived to hospital discharge, while only 66 of 75 (88.0%) adult case-controls survived to hospital discharge.

**Figure 4 animals-12-01374-f004:**
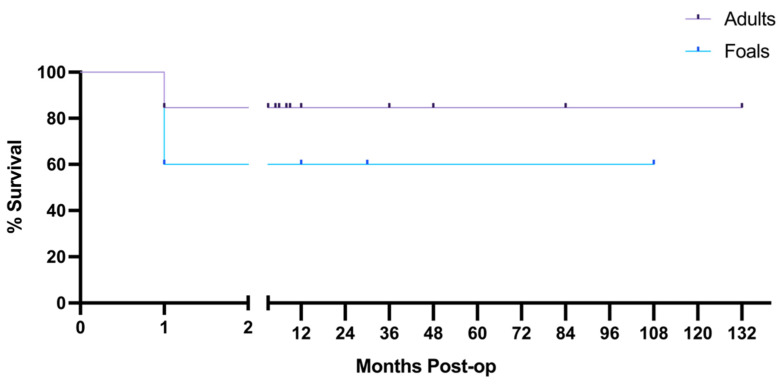
Long-term follow-up shows higher survival rates in adult case-controls. Of 13 adults and 5 foals, 2 adults and 2 foals were euthanized 1-month post-operatively for adhesions. Follow-up shows 84.62% survival in available adults, and 60% survival in available foals. The break in the *x*-axis spans time in which no horses died or were lost to follow-up. Tick marks on the survival curve indicate a horse that was lost to follow-up.

**Table 1 animals-12-01374-t001:** Univariate analyses of risk factors affecting survival.

Variable	Foals Survived (%)	Adults Survived (%)	Odds Ratio	Confidence Interval
Age	Range 1d-6m	Range 2–10y	1.005	0.9968 to 1.013
	Mean 2.18m	Mean 11.05y		

No resection	15 (93.75)	37 (88.10)	1.0	N/A
Resection	9 (100)	28 (84.85)	1.003	0.8143 to 1.148

No External Hernia	20 (95.24)	52 (85.25)	1.0	N/A
External Hernia	4 (100)	14 (100)	1.139	0.05805 to 7.332

No Volvulus	12 (100)	47 (94)	1.0	N/A
Volvulus	12 (92.31)	19 (76)	3.843	0.9964 to 18.76

Non-Thoroughbred	16 (100)	50 (90.91)	1.0	N/A
Thoroughbred	8 (88.89)	15 (75)	2.750	0.7085 to 10.71

Non-Quarter Horse	23 (95.83)	57 (90.48)	1.0	N/A
Quarter Horse	1 (100)	9 (75)	3.078	0.5978 to 13.01

## Data Availability

De-identified case data may be obtained from the corresponding author by reasonable request.

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
