# Peer review of "Multi-Institutional Retrospective Case-Control Study Evaluating Clinical Outcomes of Foals with Small Intestinal Strangulating Obstruction: 2000–2020"

_animals, 2022, doi:10.3390/ani12111374_

Round 1
Reviewer 1 Report
This MS reports surgical colic cases from foals with an attempt to match the cases with similar adult cases. Unfortunately small number of foals different pathophysiology behind the foal and adult cases makes the statistical approach chosen impossible. Therefore conclusions can not be drawn. Long term survival can not be assessed with such patchy data.
More specific comments for improvement:
-Try not to repeat results in text and table
-Present only figures that contain new data (eg. 3a vs 3b)
-Mention the statistical tests in M&M section, not with every result
-Decide if you want to focus on sociology or veterinary medicine
-Institutional ethical approval should be obtained
Author Response
Dear Reviewer,
We sincerely thank you for your time, careful consideration, and comments. We acknowledge and are grateful for your effort, and have done our best to mirror this effort in our revisions and responses to your concerns. Please see our detailed, point-by-point responses below:
Point 1: Unfortunately small number of foals different pathophysiology behind the foal and adult cases makes the statistical approach chosen impossible. Therefore conclusions cannot be drawn.
As this is a retrospective case-control study, it is designed to help determine if an attribute is associated with an outcome (here the attribute is young age). We agree that these cases are difficult to match due to the differing pathophysiology encountered in the two age groups, and so we attempted this matching with great care and consideration. We identified the cases (foals) and the controls (adults). Then we identified which subjects in each group had the outcome (here the outcome would be death or euthanasia prior to discharge from the hospital), comparing the frequency of the outcome in the case group to the control group (ISBN 978-0-19-531450-2). Indeed, in this manuscript, we selected a group of foals with strangulating lesions, identified adults with comparable strangulating lesions, and compared short term survival rates between groups. The small number of foals is due to the rarity of foal strangulating lesions which are taken to surgery, at least at the institutions studied. The limited numbers led us to use the Fisher’s Exact Test, rather than the Chi-Square which is used with larger sample numbers.
Point 2: Long term survival cannot be assessed with such patchy data.
Thank you for this comment, we acknowledge that our long-term follow-up data is patchy. We pursued long-term follow up for every case we included in the study, but unfortunately this proved impossible. Horses change locations and owners more frequently than other pets and livestock, owner contact information changes, and some institutions do not routinely gather follow-up after cases are discharged from the hospital. We did think it important to include what data we could collect, and we now state explicitly that these data are under powered. We maintain that this information is important to the manuscript as it is clinically interesting but have adjusted the manuscript to reflect this lack of statistical significance more clearly.
Point 3: More specific comments for improvement:
-Try not to repeat results in text and table
Thank you for this comment, we have tried to reduce redundancy in the manuscript, and hope this effort has been sufficient. Please provide specific instances of redundancy you would like resolved if needed, we are happy to condense more.
-Present only figures that contain new data (eg. 3a vs 3b)
Thank you for this suggestion, and we see your point regarding presentation of these data. We have selected only one set of graphs to present in Figure 3.
-Mention the statistical tests in M&M section, not with every result
Thank you for this advice, we agree that removing the statistical test names from the results helped the section read more smoothly.
-Decide if you want to focus on sociology or veterinary medicine
Thank you for this interesting suggestion, we will consider this further. We have added a brief paragraph into the discussion explaining the link between the broader sociological context and the implications for this specific study, hopefully that helps to clarify some more.
-Institutional ethical approval should be obtained
Thank you for ensuring we have met all ethical obligations to the animals and clients included in this study. This is indeed the most important aspect of any study in veterinary medicine. We pursued IACUC approval before beginning this study, and the NC State University IACUC office stated that approval is not required for retrospective studies since no live animals are involved in the research project, the case records are historical information, and no experimental interventions were applied. Similarly, because all data collected were de-identified, we did not record any human data that would be identifiable, so IRB approval was deemed not needed by the NC State University IRB office. Following your suggestion here, I verified with both offices that ethical approval is not required for this type of study.
Reviewer 2 Report
Dear authors !
The paper compares with a multi-institutional retrospective case-control study the outcome of strangulating small intestinal lesions in foals and in adult horses. It is of interest as few data and no recent ones are available about survival in foals compared to adults. Generally the study is well designed. The choice of the lesions in the adult control horses does not seem completely adequate (eventration). The conclusion goes too far, as you do not have sufficient data for longterm survival and so you should limit your conclusions to short-term survival.
Please find some other specific comments and recommendations in thefollowing:
Introduction:
reference for survival rate in foals is published in 1996 and takes cases from 1980 on, so it’s difficult to compare to more recent papers in adults. This re-enforces the utility of your study.
MM
Line 90-91 Explain better what you did with the cases euthanized intra-op (add “but analysed separately), as you say that you select 3 adult horses recovering from anaesthesia for one foal ? (with the flow-chart it’s easy to understand)
Results
Line 160 Richter’s hernia is a term used for incomplete incarceration of an intestine (only anti-mesenteric part of the intestinal wall) and so you should specify if it’s umbilical Richter’s hernia or another type of Richters hernia?
Line 163: explain difference between gastrosplenic ligament rents and gastrosplenic entrapments (it is the same for me?). If there is only a rent without incarceration, this is not a problem and how to you have an entrapment without a rent (only blocked like the colon between the spleen and the stomach? Please specify or correct and give the same diagnosis for both type of cases (perhaps in different clinics they use different terms?) The most frequently used term to me seems to be incarceration or entrapment of small intestine into the gastrosplenic ligament (ISIGL or ESIGL) (Jenei et al., 2007; Bergren et al., 2015)
Line 164 Why did you include post-castration eventration? This is not only a strangulating obstruction (like inguinal hernia) as the intestine is in contact with dirt, hair coat etc. and so for me it is not the same type of lesion than all the others. Generally, survival rates are lower than for strangulating obstruction and only range from 36-87 % in the published studies (Thomas et al., 1998). I would propose to exclude these cases and find something more similar to the other lesions, if possible. Even epiploic foramen entrapment seems more similar to other incarcerations as eventration. Not all studies show such a low survival rate for EFE.
Line 165 A question not really related with the review, but of interest for me, how did a horse develop a rent in the nephro-splenic ligament?
Line 183 Long-term follow-up based on a very small number of cases (only 20 % of the cases), therefore you should be very cautious with your conclusions
Line 195 what do you include in the lesion type “hernia”? Where are the results of the multivariate analysis? Is it possible to have significant parameters in multivariate analysis if there is no significant parameter in univariate analysis?
Discussion
You should add a small paragraph on the survival rate long-term and explain that you cannot draw conclusions as the follow-up is very limited (only 20 % of cases) (not only at the end of the discussion). Possibly, long term survival is much lower in foals compared to adults and so you should more clearly state that short term survival in foals is similar to adults.
Line 245-247
Concerning ascarid impaction, you compared this to ileal impaction with food but in your discussion you compare to strangulating lipoma ? Furthermore, until recently, ascarid impaction was associated with a low survival rate.
Conclusions : you should more cautious and state that short-term outcome of surgically treated SISO foals is similar to adults
Author Response
Dear Reviewer,
We sincerely thank you for your time, careful consideration, and comments. We acknowledge and are grateful for your effort, and have done our best to mirror this effort in our revisions and responses to your concerns. Please see our detailed, point-by-point responses below:
Introduction:
reference for survival rate in foals is published in 1996 and takes cases from 1980 on, so it’s difficult to compare to more recent papers in adults. This re-enforces the utility of your study.
Thank you for this comment acknowledging the utility of this study, this is appreciated!
MM
Line 90-91 Explain better what you did with the cases euthanized intra-op (add “but analysed separately), as you say that you select 3 adult horses recovering from anaesthesia for one foal ? (with the flow-chart it’s easy to understand)
Thank you for pointing out that this area can be further clarified. We have reworded this section in the manuscript in hopes of making this information easier to find.
Results
Line 160 Richter’s hernia is a term used for incomplete incarceration of an intestine (only anti-mesenteric part of the intestinal wall) and so you should specify if it’s umbilical Richter’s hernia or another type of Richters hernia?
Thank you for identifying this area where we should’ve been more clear. These were umbilical Richter’s hernias, and this has been updated and clarified in the manuscript.
Line 163: explain difference between gastrosplenic ligament rents and gastrosplenic entrapments (it is the same for me?). If there is only a rent without incarceration, this is not a problem and how to you have an entrapment without a rent (only blocked like the colon between the spleen and the stomach? Please specify or correct and give the same diagnosis for both type of cases (perhaps in different clinics they use different terms?) The most frequently used term to me seems to be incarceration or entrapment of small intestine into the gastrosplenic ligament (ISIGL or ESIGL) (Jenei et al., 2007; Bergren et al., 2015)
We appreciate you pointing this out; we believe this is a case of different terms being used at different institutions for the same lesion, and this has been clarified as gastrosplenic ligament entrapment in the manuscript.
Line 164 Why did you include post-castration eventration? This is not only a strangulating obstruction (like inguinal hernia) as the intestine is in contact with dirt, hair coat etc. and so for me it is not the same type of lesion than all the others. Generally, survival rates are lower than for strangulating obstruction and only range from 36-87 % in the published studies (Thomas et al., 1998). I would propose to exclude these cases and find something more similar to the other lesions, if possible. Even epiploic foramen entrapment seems more similar to other incarcerations as eventration. Not all studies show such a low survival rate for EFE.
This is a fair point, thank you for bringing this to our attention. We agree that this is a potential confounder which can be minimized by modifying our case matching in these instances. We have replaced these cases in the manuscript and have updated the statistical results accordingly. Changing the case matching did not alter the statistical outcomes or interpretation.
Line 165 A question not really related with the review, but of interest for me, how did a horse develop a rent in the nephro-splenic ligament?
That is a great question! These details weren’t available in the case’s surgical notes, unfortunately.
Line 183 Long-term follow-up based on a very small number of cases (only 20 % of the cases), therefore you should be very cautious with your conclusions
This is a great point, we attempted to be careful here. We have reworded this further to state explicitly that these data are underpowered and while, clinically interesting, definitive conclusions cannot be drawn. We hope you find this new statement more appropriate.
Line 195 what do you include in the lesion type “hernia”? Where are the results of the multivariate analysis? Is it possible to have significant parameters in multivariate analysis if there is no significant parameter in univariate analysis?
We agree that we can be clearer regarding how we define “hernia” in this study. “Hernia” broadly is referring to external hernias, and we have updated this to be more specific and clearer in the manuscript. There are no significant results from the multiple logistic regression. This reference to the multivariate analysis has been removed from the manuscript since these data were not presented.
Discussion
You should add a small paragraph on the survival rate long-term and explain that you cannot draw conclusions as the follow-up is very limited (only 20 % of cases) (not only at the end of the discussion). Possibly, long term survival is much lower in foals compared to adults and so you should more clearly state that short term survival in foals is similar to adults.
These are great suggestions, and we have added this suggested short paragraph into the discussion, and have attempted to further clarify the rest of the discussion.
Line 245-247
Concerning ascarid impaction, you compared this to ileal impaction with food but in your discussion you compare to strangulating lipoma ? Furthermore, until recently, ascarid impaction was associated with a low survival rate.
We agree that this was unclear and we have attempted to clarify these points in the discussion. We make two separate points here, the first being that we acknowledge some lesions occur only in foal aged horses while others are only seen in adults (ascarid impactions in foals, and strangulating lipomas in adults). Secondly, some lesions only seen in one age group (here the example is the ischemic injury secondary to an ascarid impaction) have comparable lesions seen in adults (ischemic injury secondary to a feed impaction). I hope we have successfully delineated between and clarified these two individual points in the manuscript.
Conclusions : you should more cautious and state that short-term outcome of surgically treated SISO foals is similar to adults
We agree and we have changed the wording here in an effort to make this more explicit: that we are only making conclusions about short-term outcome.
Reviewer 3 Report
Manuscript Number: animals-1679068
Title: Multi-institutional Retrospective Case-control Study Evaluating Clinical Outcomes of Foals with Small Intestinal Strangulating Obstruction: 2000-2020.
General comment
The present retrospective study evaluated clinical outcome of foal population after a SISO. It compared outcomes of a population of foals, referred to 5 different clinics, with a population of adult horses case-matched. The study showed that the short-term survival tended to be higher in foals than in adults. The work showed few incongruences that should be addressed.
Materials and Methods
Line 91-92: It is not fully understandable the present sentence. Could you explain it in different words? Did you mean that each adult horse has match with each foal case? Please give details.
Lines 98.100: Would it be any discrepancy due to medical complications related to ascarid impaction in foal compared with a food material impaction in adults? Please give details and add references.
Lines 101-103: could you give an explanation for did not match the euthanized ones? Would it be worthy to analyse euthanized population cases and give statistical support to your considerations? Please give details.
Results
Population trend analysis might be described just as speculation, since there was not performed statistical analysis comparing the different age distributions. Please, rephrase the sentences.
Lines 142-154: it would be useful to summarize results obtained for breed distribution in a table that underline the comparison between foal and adults who survived the surgery, in relation to their breed.
Discussion
There is extensive discussion about the reasons for the decision to refer, in case of severe colic signs. It might be worthy to focus the paper discussion more on the outcome of foals referred for SISO compared to case-matched adults, and the survival rate after colic surgery.
Which were the main reasons for intraoperative euthanasia? Could the future quality of life or expected performance have influenced the decision-making process? Which might be the differences between foals and the adult's population? Please give details.
Author Response
Dear Reviewer,
We sincerely thank you for your time, careful consideration, and comments. We acknowledge and are grateful for your effort, and have done our best to mirror this effort in our revisions and responses to your concerns. Please see our detailed, point-by-point responses below:
Materials and Methods
Line 91-92: It is not fully understandable the present sentence. Could you explain it in different words? Did you mean that each adult horse has match with each foal case? Please give details.
Thank you for highlighting where we can be clearer. We have attempted to further clarify this section of the manuscript. We collected foal cases first, and then matched three adult cases to each foal case that was recovered from anesthesia.
Lines 98.100: Would it be any discrepancy due to medical complications related to ascarid impaction in foal compared with a food material impaction in adults? Please give details and add references.
While we accept that ascarid impactions are inherently more severe than feed impactions, care was taken in selections of adult case matches. Surgical notes were referenced to verify similar severity in damage between lesions. We hope that this has been sufficiently clarified in the manuscript.
Lines 101-103: could you give an explanation for did not match the euthanized ones? Would it be worthy to analyse euthanized population cases and give statistical support to your considerations? Please give details.
Thank you for this suggestion. The main reason we collected cases euthanized intraoperatively was to screen for any bias on the part of the surgeon against foal cases. When we found that similar proportions of foals and adults were euthanized in surgery, we reported these finding, and decided to case match recovered foals and further study their postoperative clinical outcomes. From here, we proceeded with assessment of short-term survival and pursuit of long-term follow up. We hope this helps clarify our rationale.
Results
Population trend analysis might be described just as speculation, since there was not performed statistical analysis comparing the different age distributions. Please, rephrase the sentences.
Our interpretations of the population distributions were driven by results of normality testing. When the age distributions were found not to be normally distributed, we described the distribution of the data based on the appearance of the histogram. We agree that this can be further clarified with more objective data, and so we have updated the manuscript to report mean and median data for each data set.
Lines 142-154: it would be useful to summarize results obtained for breed distribution in a table that underline the comparison between foal and adults who survived the surgery, in relation to their breed.
We agree this could be useful information for many readers, and so we have compiled a supplemental table providing this information.
Discussion
There is extensive discussion about the reasons for the decision to refer, in case of severe colic signs. It might be worthy to focus the paper discussion more on the outcome of foals referred for SISO compared to case-matched adults, and the survival rate after colic surgery.
Thank you for this suggestion, we have added a brief paragraph into the discussion explaining the link between the broader sociological context and the implications for this specific study (starting at line number 213), along with further clarifying the discussion broadly. Hopefully these adjustments help to clarify some more.
Which were the main reasons for intraoperative euthanasia? Could the future quality of life or expected performance have influenced the decision-making process? Which might be the differences between foals and the adult's population? Please give details.
Thank you for inquiring about this, unfortunately the reason for euthanasia was infrequently present in the surgical and case notes available to us. This would be a great variable to measure and control for in a future study.
Round 2
Reviewer 1 Report
Point 1: As the supplemetary table shows, it is not possible to do case matching in a pathophysiologically meaningful way. You will end up comparing apples to peaches no matter what you do. For example intussusception is a disease that affects foals and any attempt to compare it with adult diseases is not meaningful. Therefore I recommend, that you publish your results (maybe both foals and adults) as a clinical report from your hospitals without trying to make statistical comparison between foals and adults (other than descriptive).
Since 24/25 survival rate is strikingly different from any previous reports or clinical experience, some explanation (other than euthanasing hopeless cases during surgery) is much needed.
Point 3:
Chapter 3.1. describes the normality of the data, which is usually of interest only when choosing appropriate statistical testing. It does not need to be presented as a result. Figure 1 shows already to the reader, how the data is skewed.
Figure 2:
The data of figures2a and 2c is included in figure 2b and 2d.
The same text is found in duplicate in lines 118-131 and 134-146. Remove all results from the caption. The caption should only contain necessary information for the reader to understand the figure. Also, remove the word "affects" from line 133, since no interpretation should be included in the caption.
Figure 3: Remove all the result from caption, they are already mentioned in the text. Lines 180-181: remove interpretation.
Figure 4: Remove replication of results from the caption (as before).
Line 128: Replace & with some more appropriate edition.
Line 178-179: P=0.4 is not a trend.
Table 1: Check excessive use of desimals
Line 203: You can not say higher, when there is not statistical significance.
Line 225: I would not consider this very recent
Reviewer 2 Report
I feel satisfied with the corrections of the paper. The authors could respond to all my remarks and questions.
Reviewer 3 Report
the paper might be considered suitable for publication in the present form.